# Outcome after Resection for Hepatocellular Carcinoma in Noncirrhotic Liver—A Single Centre Study

**DOI:** 10.3390/jcm11195802

**Published:** 2022-09-30

**Authors:** Lea Penzkofer, Jens Mittler, Stefan Heinrich, Nicolas Wachter, Beate K. Straub, Roman Kloeckner, Fabian Stoehr, Simon Johannes Gairing, Fabian Bartsch, Hauke Lang

**Affiliations:** 1Department of General, Visceral and Transplant Surgery, University Medical Center of the Johannes Gutenberg University Mainz, 55131 Mainz, Germany; 2Institute of Pathology, University Medical Center of the Johannes Gutenberg University Mainz, 55131 Mainz, Germany; 3Department of Diagnostic and Interventional Radiology, University Medical Center of the Johannes Gutenberg University Mainz, 55131 Mainz, Germany; 4Department of Internal Medicine I, University Medical Center of the Johannes Gutenberg University Mainz, 55131 Mainz, Germany

**Keywords:** HCC, non-cirrhotic liver, liver resection, overall survival, intrahepatic recurrence-free survival

## Abstract

Liver cirrhosis is the most common risk factor for the development of hepatocellular carcinoma (HCC). However, 10 to 15% of all HCC arise in a non-cirrhotic liver. Few reliable data exist on outcome after liver resection in a non-cirrhotic liver. The aim of this single-centre study was to evaluate the outcome of resection for HCC in non-cirrhotic liver (NC-HCC) and to determine prognostic factors for overall (OS) and intrahepatic recurrence-free (RFS) survival. From 2008 to 2020, a total of 249 patients were enrolled in this retrospective study. Primary outcome was OS and RFS. Radiological and pathological findings, such as tumour size, number of nodules, Tumour-, Nodes-, Metastases- (TNM) classification and vascular invasion as well as extent of surgical resection and laboratory liver function were collected. Here, 249 patients underwent liver resection for NC-HCC. In this case, 50% of patients underwent major liver resection, perioperative mortality was 6.4%. Median OS was 35.4 months (range 1–151 months), median RFS was 10.5 months (range 1–128 moths). Tumour diameter greater than three centimetres, multifocal tumour disease, vascular invasion, preoperative low albumin and increased alpha-fetoprotein (AFP) values were associated with significantly worse OS. Our study shows that resection for NC-HCC is an acceptable treatment approach with comparatively good outcome even in extensive tumours.

## 1. Introduction

Hepatocellular carcinoma (HCC) is the sixth most common tumour and the most common primary liver tumour, accounting for over 80% of all malignant primary liver tumours [1]. Liver cirrhosis is by far the most common risk factor [2,3,4]. However, approximately 10 to 15% of HCCs arise in a non-cirrhotic liver. These tumours are mainly related to non-alcoholic fatty liver disease (NAFLD) or chronic hepatitis B (HBV) virus infection [5,6,7,8]. The proportion of HCC in non-cirrhosis is supposed to increase in the future, mainly due to an increasing prevalence of NAFLD in the Western world [9,10,11,12].

Liver transplantation, liver resection and local ablation are the only curative treatment options for HCC [13]. The treatment of non-cirrhotic HCC (NC-HCC) is the domain of liver resection as no or little hepatic dysfunction of the non-tumorous liver parenchyma allows for even extended resection. Liver transplantation is only exceptionally indicated for NC-HCCC in rare non-resectable primary or recurrent tumours. Local ablation is limited to early stage HCC and therefore is no option in most cases, too [14,15].

In recent years, significant progress has been made in both the diagnosis and treatment of HCC. However, the majority of studies does not distinguish between resection for HCC in cirrhotic or non-cirrhotic liver. Hence, there are few reliable data on the outcome after liver resection for NC-HCC.

The aim of this single-centre study was to evaluate the outcomes of resections for NC-HCC and to determine prognostic factors for overall and intrahepatic recurrence-free survival (OS, RFS).

## 2. Materials and Methods

The data from all consecutive patients who underwent liver resection for NC-HCC between January 2008 and December 2020 were prospectively collected in an institutional database. Data were subsequently transferred to SPSS (IBM SPSS Statistics for Windows, Version 26, IBM, Armonk, NY, USA) for further analysis. Absence of liver cirrhosis was proven by a complete pathological examination of the resected specimen. The classification of fibrosis was made according to Desmet et al. [16].

Paediatric patients (age < 16 years) and patients with mixed hepato- and cholangiocellular carcinoma were excluded from analysis. One patient with previous history of intrahepatic cholangiocellular carcinoma was excluded as well. A total of 249 patients finally met the inclusion criteria.

Preoperative work-up included an abdominal computer tomography (CT) or magnetic resonance imaging (MRI) scan as well as a chest radiography or CT scan of the thorax. Intraoperative ultrasound was performed as part of the standard procedure. Liver function was assessed preoperatively by means of bilirubin, albumin and quick value. Alpha-feto-protein (AFP) values were recorded.

All cases were discussed, and all treatment decisions were made by a multi-disciplinary tumour board. In general, patients with distant metastases were excluded from surgical therapy. In an individual therapeutic approach, surgery was however indicated by a case-by-case decision in very young patients with limited extrahepatic spread. In these cases, surgery was part of a multimodal therapeutic concept. Neither advanced T nor N stage were exclusion criteria. Only patients with unimpaired liver function qualified for resection and those with signs of portal hypertension such as splenomegaly, intraabdominal varices or ascites were no candidates for resection. If extended resections were necessary additional volumetry of the future liver remnant was performed by an experienced radiologist. Liver resection was initially recorded according to Couinaud’s segment classification [17], since 2020 according to the ‘New World’ Terminology [18]. A non-anatomic resection or the anatomic resection of one to two segments was defined as minor resection, an anatomic resection of three or more segments as major resection. Trisectionectomies and mesohepatectomies were registered separately. Histopathological information was recorded on the basis of the current Tumour-, Nodes-, Metastases- (TNM) classification [19].

Follow-up examinations included imaging examinations every three months in the first year after surgery. Ultrasound and CT scans were performed alternately. Afterwards, examinations took place every 6 to 12 months. For patients whose follow-up examinations were not carried out at our clinic, the physicians who continued their treatment were contacted.

Categorical data were calculated using the Chi-square test. Connected samples were analysed using the Wilcoxon signed-rank test, for independent samples a *t*-test was used. Univariate survival analyses were performed using Kaplan-Meier curves with log-rank test. For calculation of OS and RFS, the day of liver resection served as baseline. RFS was calculated according to Punt and colleagues [20]. Perioperative deaths were excluded from survival analyses. Cox regression (proportional hazards model) with backwards selection was used for multivariate analyses. *p*-values < 0.05 were considered significant.

## 3. Results

### 3.1. Baseline Characteristics

A total of 249 patients underwent liver resection for NC-HCC. Baseline characteristics are summarised in Table 1. The majority of patients had no known underlying liver disease. The most common underlying liver diseases were viral hepatitis (18%), NAFLD (14%) and alcoholic liver disease (ALD, 12%). One patient had autoimmune hepatitis and glycogenosis, respectively.

In this case, 12 patients had undergone preoperative treatment of HCC with transarterial chemoembolization (TACE, *n* = 5), systemic chemotherapy (*n* = 3), ablation by radiofrequency (RFA, *n* = 1), TACE and RFA (*n* = 2) and TACE followed by systemic chemotherapy (*n* = 1).

### 3.2. Liver Funcion

On median, all assessed liver function parameters (albumin, bilirubin, quick) were within the normal range preoperatively.

Alpha-fetoprotein serum levels (AFP, cut off value: 8.8 ng/mL) were available in 211 patients. A total of 110 patients had an AFP serum level within the normal range (median 3.2 ng/mL), 101 patients had elevated AFP serum levels with a median of 256 ng/mL (range 8.9 to 572,734 ng/mL); of these, 20 had an AFP value in the five- to six-digit range.

### 3.3. Preoperative Imaging

Preoperative imaging showed a unifocal tumour in 76% (*n* = 189) of all patients. However, 12% (*n* = 29) of patients had multifocal tumour disease with four or more lesions (Table 2). Taking all patients into account, the median diameter of the largest lesion was 70 mm (range 10 to 300 mm). Portal vein invasion was suspected in 13 cases (5.2%) with another 12 cases (4.8%) in which the main branches of the portal vein could not be evaluated properly on imaging.

### 3.4. Surgical Procedures

Table 3 provides an overview of the surgical procedures. 50% of patients (*n* = 125) underwent major liver resection. Trisectionectomy was performed in 36 cases (14.5%) and mesohepatectomy in nine cases (3.6%). Lymphadenectomy in the hepatic hilus was undertaken in 41% of patients (*n* = 103). A median of two lymph nodes (range 1–18) was resected.

Additional extrahepatic resection was performed in 43 cases (17.3%), including partial resection of the diaphragm (*n* = 18, 7.2%), right adrenalectomy (*n* = 6, 2.4%), extended lymphadenectomy intrathoracal, paraaortic, or retropancreatic (*n* = 5, 2.0%), small or large bowl resection (*n* = 4, 1.6%), tumour resection on the peritoneum or the bursa omentalis (*n* = 4, 1.6%), nephrectomy (*n* = 2, 0.8%), bile duct resection with biliodigestive anastomosis (*n* = 1, 0.4%), resection of an inoculation metastasis after liver biopsy (*n* = 1, 0.4%) and thrombectomy from the right atrium (*n* = 1, 0.4%). One patient (0.4%) underwent simultaneous gastric repositioning with fundopexy.

### 3.5. Mortality

A total of 16 patients (6.4%) died within hospital stay. Causes of death were cardiopulmonary failure (*n* = 6, 2.4%), acute liver failure (*n* = 5, 2.0%) and sepsis leading to multi organ failure (*n* = 4, 1.6%). One patient (0.4%) died during the inpatient stay after surgery due to a traumatic femoral neck fracture. All patients who died of acute liver failure had received major liver resection, three patients had undergone hemihepatectomy, two patients trisectionectomy.

### 3.6. Pathological Examination

Table 4 shows a summary of the pathological results. 197 patients (79.1%) had liver fibrosis grade I or higher according to Desmet’s grading.

In 216 of 249 patients (86.7%) a R0 resection with a median resection margin of 2 mm (range 1 to 50 mm) was performed. A R1 resection was performed in 17 patients (6.8%), four of which were classified as R1 resection due to previous tumour rupture. Furthermore, four patients had an R1 vascular situation. R2 resection was performed in nine cases (3.6%). In seven patients (2.8%), the resection margin could not be clearly assessed due to tissue fragmentation, leading to an Rx situation.

In six of the nine R2 resections, unfavourably located lesions were left in situ for further therapy with TACE or RFA. In another two cases, debulking surgery was performed in an individual approach as the patients were symptomatic due to high tumour burden. One case was classified as R2 resection because of a tumour thrombus in a major hepatic vessel containing vital tumour cells.

Hilar lymph node metastases were present in 7 patients (2.8%), all of whom had underwent major liver resection and additional extrahepatic surgery in four cases.

Vascular infiltration was found in 116 patients (46.5%). 36% (*n* = 90) had microvascular infiltration corresponding to a V1 stage. 10% (*n* = 26) had a V2 stage in the sense of macrovascular tumour invasion. In 50% of these cases (*n* = 13), the portal vein was infiltrated. In the remaining cases, the main hepatic veins (3.6%, *n* = 9) or the inferior vena cava (1.6%, *n* = 4) were affected.

Distant metastasis (M1 stage) was present in 15 patients (6.0%). In four cases, distant metastasis was an intraoperative incidental finding. In 11 cases, distant metastasis was known preoperatively. Surgery was performed as part of an individual therapy concept, for example, in cases of symptoms due to an extensive tumour burden in the liver (*n* = 4) or as a curative intended concept in very young patients (*n* = 4). In most cases, the diaphragm (*n* = 3) or the lungs (*n* = 2) were affected. One patient each had simultaneous metastases to the lung and diaphragm, and to the diaphragm and right colonic flexure, respectively. The remaining cases involved metastases in the right adrenal gland (*n* = 2), distant lymph node metastases (*n* = 2), bone (*n* = 1), peritoneum (*n* = 1), omentum majus (*n* = 1), and a singular abdominal skin metastasis (*n* = 1). With the exception of one patient with a bone metastasis to the os ilium and three patients with a metastasis to the lung, all distant metastases were resected during liver surgery.

### 3.7. Interacting Parameters

#### 3.7.1. AFP

Patients with an elevated AFP level were more likely to have an advanced T stage as well as poorly differentiated tumours in pathological examination. They suffered more frequently from tumours of more than 50 mm in diameter. The rate of vascular invasion was significantly increased (*p* < 0.001, respectively). No difference was found with regard to the required extent of resection.

#### 3.7.2. Vascular Infiltration

Patients with vascular infiltration were more likely to have advanced tumour stage (*p* < 0.001) and poorly differentiated tumours (*p* = 0.003). Additionally, they required extensive resection more frequently (*p* < 0.001).

#### 3.7.3. Tumour Diameter

Patients with a maximum tumour diameter of 50 mm or more in the preoperative radiological workup or the pathological examination presented with more advanced T stages and a higher rate of vascular invasion and lymph node metastases (*p* < 0.001, respectively). There was no difference regarding the differentiation of HCC and the grade of liver fibrosis

### 3.8. Survival

Median OS was 35.4 months ranging between 1 and 151 months. The 1-, 3-, and 5-year overall survival rates were 84%, 64%, and 47%, respectively. Median intrahepatic RFS was 10.5 months ranging between 1 and 128 months. The 1-, 3-, and 5-year intrahepatic RFS rates were 46%, 20%, and 7%, respectively.

Patients with a R0 resection showed a significantly longer OS than patients with an R1 and R2 resection (59.0 months vs. 16.3 months, *p* = 0.003, cf. Table 5, Figure 1).

OS differed significantly across all tumour stages. Patients with a very early tumour stage (T1) showed the longest median OS with 85.0 months. In advanced tumour stage (T4), the median OS decreased to 9.5 months (cf. Table 5, Figure 2).

Vascular invasion was significantly associated with worse OS across all subgroups. Patients without vascular invasion showed the longest OS with 72.7 months. With macroscopic invasion, the median OS was reduced to 19.6 months (cf. Table 5, Figure 3).

#### 3.8.1. Univariate Analyses of Survival

Univariate Kaplan-Meier analyses were used to investigate potential factors influencing OS and intrahepatic RFS. Preoperative imaging, surgical procedure, pathologic examinations, demographic aspects and laboratory chemistry results were considered.

For OS tumour size and portal vein invasion, both in the preoperative imaging and the pathological examination had significant impact. Additionally, numbers of lesions in total, a unifocal tumour disease as well as multifocality were found as influencing factors. In pathological examination tumour stage, grading, resection stage and vascular infiltration showed *p* values below 0.05. Extend of liver resection, additional extrahepatic surgery, ASA Classification and preoperatively assessed AFP value had significant impact on OS. A subgroup analysis of 51 patients with a preoperative albumin below the normal range also showed a worse OS for these patients.

Regarding RFS a unifocal lesion and multifocal tumour disease in preoperative imaging as well as in pathological examination had significant influence. Vascular invasion and portal invasion in particular were associated with RFS as well. Sex and extrahepatic resection also showed significant influence. Among preoperative assessed laboratory values only AFP was associated with RFS. Liver fibrosis as well as the presence or absence of liver disease showed no influence neither on OS nor on RFS.

#### 3.8.2. Multivariate Analyses of Survival

All factors that had shown an influence on OS and RFS in the univariate analyses were included in a multivariate Cox regression model using backward selection.

Tumour size of more than 3 cm in preoperative imaging (HR = 0.190; 95% CI: 0.041–0.878; *p* = 0.034) and in pathological examination (HR = 5.951; 95% CI: 1.175–30.150; *p* = 0.031) were associated with OS. A unifocal tumour disease (HR = 1.775; 95% CI: 1.042–3.022; *p* = 0.035), vascular infiltration (HR = 1.626; 95% CI: 1.093–2.418; *p* = 0.016) and preoperative albumin value (HR = 0.414; 95% CI: 0.241–0.712; *p* = 0.001) as well as AFP value (HR = 1.718; 95% CI: 1.016–2.906; *p* = 0.043) also had significant impact on OS.

Vascular infiltration (HR = 1.643; 95% CI: 1.232–2.191; *p* = 0.001) was negatively associated with RFS, a unifocal tumour disease in pathological examination (HR = 2.685; 95% CI: 1.569–4.593; *p* < 0.001) was positively associated with RFS. Furthermore, multifocal tumour disease in preoperative imaging (HR = 0.529; 95% CI: 0.293–0.954; *p* = 0.034) and sex (HR = 1.599; 95% CI: 1.004–2.546; *p* = 0.048) showed significant influence on RFS.

#### 3.8.3. OS and RFS in Dependence of Pathological Examination

OS was different across all tumour stages. Patients with R0 resection, early tumour stage (T1) and lack of vascular invasion (V0) had the longest OS. Patients with detectable residual tumour (R1/R2), an advanced tumour stage (T4) or macrovascular invasion (V2) had the lowest median OS (cf. Table 6).

A unifocal HCC and vascular invasion showed a significant impact on RFS. Patients without vascular invasion (V0) had a median RFS of 18 months. However, with macroscopic invasion (V2), the RFS decreased to about half a year (cf. Table 7).

#### 3.8.4. OS and RFS in Dependence of Preoperative AFP Value

The preoperatively determined AFP showed a significant influence on OS (cf. Figure 4). Patients with an AFP within the normal value (cut-off 8.8 ng/mL) had a median OS of 67 months, with an AFP above the normal value the OS decreased to 38 months.

Patients with an elevated AFP also had a shorter RFS (16 months vs. 10 months, *p* = 0.040).

## 4. Discussion

We report on a single centre study with 249 patients who underwent liver resection for NC-HCC. As only a few studies selectively address the entity of NC-HCC [21,22,23,24,25,26] this is, to the best of our knowledge, the largest surgical NC-HCC collective in the Western world. Our aim was a structured evaluation of our cohort and the determination of prognostic factors for OS and intrahepatic RFS.

Median OS was 35 months with 1-, 3-, and 5-year overall survival rates of 84%, 64%, and 47%. Multivariate analysis demonstrated that patients with a tumour diameter of less than 30 mm, a unifocal disease, lacking vascular infiltration and AFP serum levels within the normal range had a significantly better OS. Median RFS was 11 months with 1-, 3-, and 5-year RFS rates of 46%, 20%, and 7%. Longer RFS was linked as well to unifocal tumours without vascular infiltration. Patients with elevated AFP had worse OS and RFS.

Aetiology and, accordingly, the proportion of NC-HCC varies greatly by geographical region. Looking particularly at the developed countries, NAFLD is the most common underlying liver disease [27,28,29]. Soaring rates of the metabolic syndrome have led to a sharp increase in NAFLD, which ultimately results in a higher proportion of NC-HCC [30,31]. These results were confirmed in our cohort. The most frequent underlying liver disease was NAFLD, followed by ALD. In the literature, there is a high proportion of unknown causes for NC-HCC, ranging up to 57% of all cases [24,32]. In our study as well, no liver disease could be identified by history taking, clinical examination and pathological work-up in 53% of all patients.

Introduction of screening programmes for patients with chronic liver disease not yet diagnosed with liver cirrhosis is repeatedly discussed in the literature [33,34,35]. The current guidelines also indicate a great need for research to establish a clear approach [13]. Due to the high prevalence of NAFLD, a universal surveillance program is not economically efficient and therefore not established. Due to the late onset of symptoms and the lack of surveillance programmes, patients with NC-HCC frequently present with advanced tumour stage at diagnosis. Larger tumour size often requires extensive resections to achieve R0 resection [36,37,38]. These findings are consistent with the results of our study. With a median tumour diameter of 7 cm, more than 50% of patients in our cohort required major resection, almost every fifth patient underwent trisectionectomy or mesohepatectomy.

For HCC in cirrhosis, impaired liver function is the most common contraindication to surgery. Portal hypertension as well as liver function parameters are considered in the decision for or against surgery. However, clear criteria that should be considered for decision-making in NC-HCC do not exist. In a healthy liver tissue, extensive resections of up to 80% of the parenchyma are possible, which enables the high proportion of major resections [39].

Impaired liver function has been shown to be associated with a worse outcome after resection in cirrhotic liver. In particular, changes in coagulation parameters, elevated bilirubin as well as decreased albumin are indicative of preoperatively liver dysfunction. This is reflected in numerous scoring systems that are used for predicting OS after liver resection for HCC [40,41,42,43,44]. Information on how this specifically affects outcome in a non-cirrhotic liver is lacking. In the multivariate regression analysis, low albumin values showed a significant association with shortened OS in our collective. This is strong evidence that, irrespective of the presence of cirrhosis, liver function parameters should be included in the decision for or against resection. Interestingly, the aetiology of underlying liver disease had no influence on survival in our collective. Moreover, even the mere presence or absence of liver disease had no impact on either OS or RFS.

The determination of the tumour marker AFP has long been established in the screening, diagnosis and follow-up of HCC. However, the significance regarding a prognosis of OS is controversially discussed. In addition, most studies focus selectively on the relationship between AFP levels and HCC in cirrhosis. One study that compared HCC in cirrhosis and non-cirrhosis found no difference in the prognostic value of AFP [45]. A frequent criticism is that a significant proportion of HCC do not secrete AFP at all [46,47]. Furthermore, studies have shown a lack of correlation between prognostic factors such as tumour size as well as tumour number and AFP [48]. On the other hand, studies have also demonstrated an independent prognostic value of AFP on OS after resection, as well as a correlation with tumour size and number. [49,50,51]. A 50% decrease in AFP within the first six months after surgery was also associated with prolonged OS [52]. In our cohort, both OS and RFS were correlated with AFP levels. An increase in preoperative AFP above the normal value of 8.8 ng/mL was associated with a reduction in median OS from 67 to 38 months (significant in multivariate analyses); RFS was reduced from 16 to 10 months (significant only in univariate analyses). We therefore consider AFP to be a good marker to estimate OS and RFS.

Operative mortality ranges from 4 to 6.5% in most studies [53,54,55,56], which is consistent with the results of our study. Looking at the overall survival rates, they vary between 62% and 100% for 1 year, 38% and 78% for 3 years and 30% and 81% for 5 years [37]. Our overall 1-, 3-, and 5-year survival rates of 84%, 64% and 47% are in line with these findings. A more detailed analysis of the literature reveals that studies at the upper end had highly selective inclusion criteria. In one study, only HCC with very limited tumour number and size were included [57]; in another study, the higher survival rates were likewise found in a subgroup differentiated by tumour size [54]. A third study included only patients with HCC in a nonfibrotic and seronegative liver [38], whereas a fourth study as well excluded patients with underlying hepatitis [21].

OS in our collective was also higher in the subgroup with a maximum tumour diameter of 5 cm or less (96%, 85% and 64%, respectively). Early tumour stage (T1, 97%, 84% and 66%, respectively) and lack of vascular infiltration (90%, 74% and 57%, respectively) were associated with better OS compared to the studies mentioned above as well. However, we could not detect any effect of fibrosis stage on OS, which is in line with the results of Schiffman et al. [58].

It is well known that macrovascular invasion (V2) is related to poorer median OS [59,60,61]. This was equally reproducible in our cohort. Nevertheless, median OS in this particular subgroup was 20 months with a 5-year survival rate of 30%. For HCC in cirrhosis with macrovascular invasion (BCLC C) the recommended first-line therapy is Atezolizumab/Bevacizumab [62]; median survival rates vary between 4 and 8 months [63]. A recent study compares liver resection with sorafenib in HCC with macrovascular invasion. The median survival of 21 months in the resection group is in line with our results; the sorafenib group had a significantly worse OS of 12 months [64]. In case of HCC in cirrhosis, only parenchyma-sparing resection is usually possible due to limited liver function. However, in advanced HCC with macrovascular invasion, extensive liver resections are often necessary for curative treatment. In contrast, the preserved liver function in NC-HCC allows for extensive resections. Even though therapy recommendations for advanced NC-HCC are lacking our OS data justify the surgical approach as primary therapy in HCC even with macrovascular invasion.

Moreover, our results clearly show that even microscopic invasion (V1) is associated with shorter OS and RFS. Yet today, microvascular invasion can only be proven after a complete workup of the specimen. Preoperative detection, for example by biopsy, is not possible so far. It remains to be explored whether there exist surrogate parameters that could be used to predict microvascular invasion.

Overall, our study shows that the prognosis of HCC is influenced by a variety of histopathological and laboratory parameters such as vascular infiltration or preoperative albumin values. The question arises to what extent individual parameters known in advance should have an influence on the decision for and against resection. Multifocal tumour disease and vascular invasion were associated with worse survival in our collective. However, the currently available therapeutic alternatives such as systemic therapy with Atezolizumab/Bevacizumab or transarterial chemotherapy (TACE) are not curative therapeutic approaches compared with surgical resection.

Tumour diameter greater than three centimetres was also associated with worse OS. Yet, tumours with a maximum diameter of less than 3 centimetres are rare, especially in NC-HCC. In our collective, this was the case in 30 patients. However, most of the patients had a recurrence that had been detected early in the course of structured tumour follow-up.

## 5. Conclusions

Looking at the comparatively good OS, we consider surgical therapy of NC-HCC justified. If the possibility of R0 resection exists, even extensive tumours can be treated in curative intention. A low tumour stage, the absence of vascular infiltration and AFP values within the normal range are associated with improved overall survival.

## Figures and Tables

**Figure 1 jcm-11-05802-f001:**
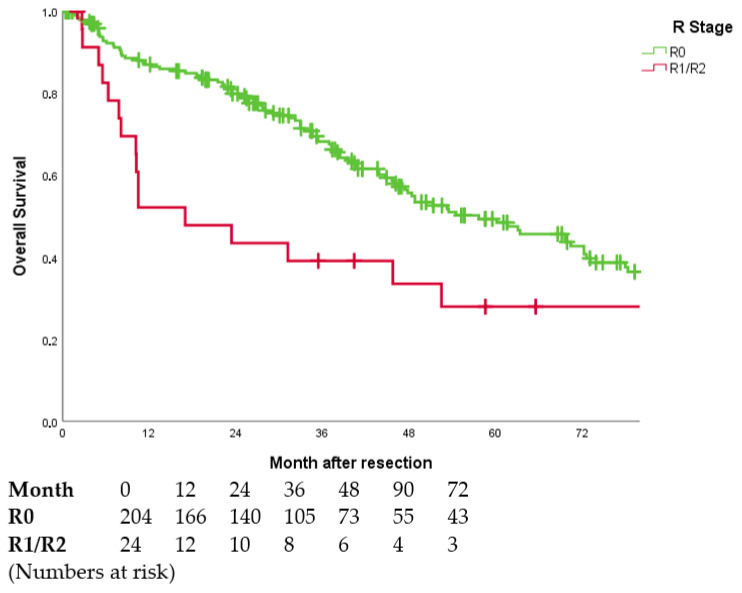
Overall survival curves in dependence of R stage (*p* = 0.003). Perioperative deaths (*n* = 16) were excluded. Five patients with unclear resection margin due to tissue fragmentation were excluded.

**Figure 2 jcm-11-05802-f002:**
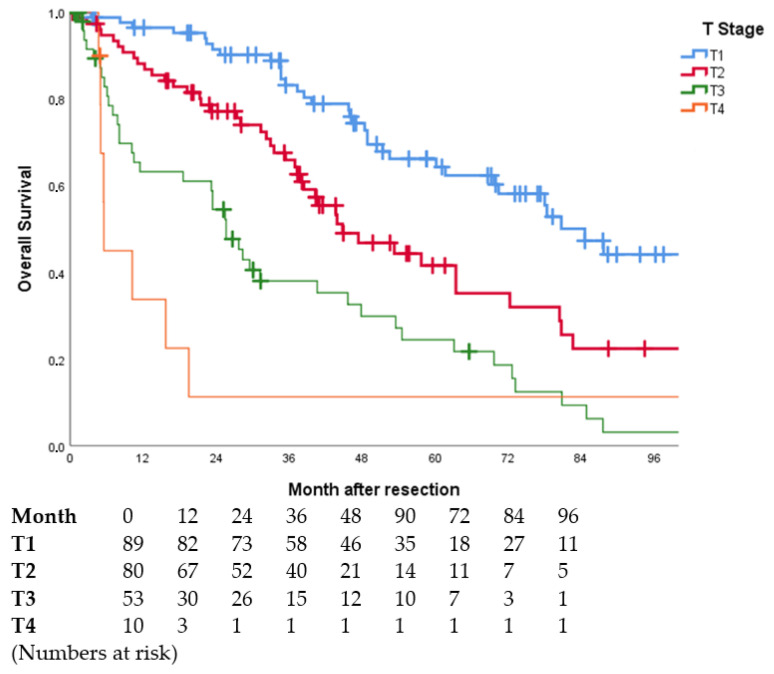
Overall survival curves in dependence of T stage (*p* < 0.001). Subgroups T1 vs. T2 (*p* = 0.001), T1 vs. T3 (*p* < 0.001), T1 vs. T4 (*p* < 0.001), T2 vs. T3 (*p* = 0.001), T2 vs. T4 (*p* < 0.001), T3 vs. T4 (*p* = 0.349). Perioperative deaths (*n* = 16) were excluded. In one patient, no information on T stage was available.

**Figure 3 jcm-11-05802-f003:**
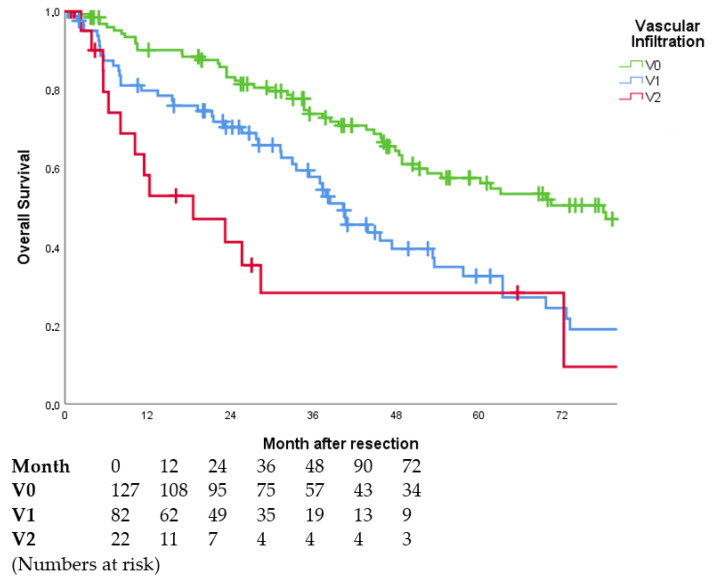
Overall survival curves in dependence of vascular invasion (*p* < 0.001). Subgroups V0 vs. V1 (*p* < 0.001), V0 vs. V2 (*p* < 0.001), V1 vs. V2 (*p* = 0.066). Perioperative deaths (*n* = 16) were excluded. In two patients, no information on vascular infiltration was available.

**Figure 4 jcm-11-05802-f004:**
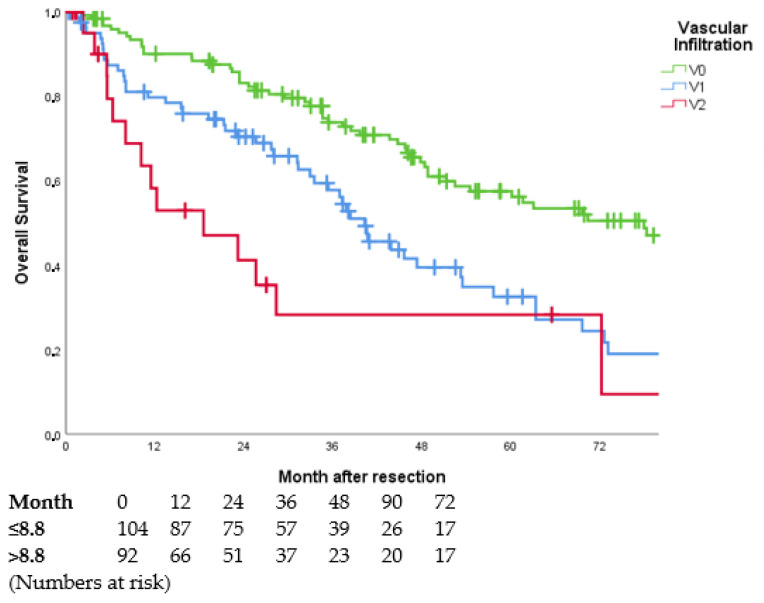
Overall survival curves in dependence of AFP value (*p* < 0.001). Perioperative deaths (*n* = 16) were excluded. AFP values were available in 196 of 233 patients. AFP: alpha-fetoprotein.

**Table 1 jcm-11-05802-t001:** Baseline characteristics (*n* = 249).

Variables	Values
Sex male/female, *n* (%)	190/59 (76.3/23.7)
Age (year, median, range)	71 (17–93)
ASA classification, *n* (%)	
I	1 (0.4)
II	67 (26.9)
III	157 (63.1)
IV	9 (3.6)
unknown	15 (6.0)
Liver disease, *n* (%)	
Unknown	133 (53.4)
NAFLD	34 (13.7)
Viral	
HBV	24 (9.6)
HCV	20 (8)
ALD	29 (11.6)
ALD and Hepatitis	1 (0.4)
Haemochromatosis	6 (2.4)
AIH	1 (0.4)
Glycogenosis	1 (0.4)
Preoperative liver function, (median, range)	
Albumin (g/L)	37 (13–46)
Total bilirubin (mg/dL)	0.60 (0.19–5.98)
Quick (%)	98 (45–135)
AFP (ng/mL)	6900 (1.1–572,734)
Outcome	
Hospitalization (days, median, range)	12 (3–126)

ASA, American Society of Anesthesiologists; NAFLD, non-alcoholic fatty liver disease; HBV, hepatitis B virus; HCV, hepatitis C virus; ALD, alcohol-related liver disease; AIH, autoimmune hepatitis; AFP, alpha-fetoprotein.

**Table 2 jcm-11-05802-t002:** Preoperative imaging.

Number of Tumours	*n* (%)
1	190 (76.3)
2	23 (9.2)
3	6 (2.4)
≥4	29 (11.6)
Unknown	1 (0.4)

**Table 3 jcm-11-05802-t003:** Preoperative imaging and surgical procedures.

Variables	Values
Number of nodules, *n* (%)	
1	190 (75.9)
2	23 (9.2)
3	6 (2.4)
≥4	29 (11.6)
Unknown	1 (0.4)
Largest nodule diameter (mm), (median, range)	70 (10–300)
Milan criteria, *n* (%)	
Yes	68 (27.3)
No	176 (70.68)
Unknown	5 (2.0)
Portal vein invasion, *n* (%)	
Yes	13 (5.2)
No	221 (88.8)
Probably	12 (4.8)
Unknown	3 (1.2)
Operative Data, *n* (%)	
Minor resection	124 (49.8)
Major resection	80 (32.1)
Trisectionectomy	36 (14.5)
Mesohepatectomy	9 (3.6)
Extrahepatic resection	43 (17.3)

**Table 4 jcm-11-05802-t004:** Pathological examination.

Variables	Values
Desmet, *n* (%)	
0	47 (18.9)
1	77 (30.9)
2	64 (25.7)
3	56 (22.5)
Unknown	5 (2.0)
Resection, *n* (%)	
R0	216 (86.7)
R1	17 (6.8)
R2	9 (3.6)
Rx	7 (2.8)
Resection margin (mm), (median, range)	2 (1–50)
T category, *n* (%)	
T1	93 (37.3)
T2	83 (33.3)
T3	57 (22.9)
T4	15 (6.0)
Unknown	1 (0.4)
N category, *n* (%)	
N0	96 (38.6)
N1	7 (2.8)
Nx	146 (58.6)
M category	
M0	234 (94.0)
M1	15 (6.0)
Tumour Grading, *n* (%)	
G1	16 (6.4)
G2	147 (59.0)
G3	75 (30.1)
G4	6 (2.4)
Unknown	5 (2.0)
Vascular Invasion, *n* (%)	
V0	131 (52.6)
V1	90 (36.1)
V2	26 (10.4)
Unknown	2 (0.8)
Portal vein invasion, *n* (%)	13 (5.2)
Vena cava invasion, *n* (%)	4 (1.6)
Main hepatic vein invasion, *n* (%)	9 (3.6)
Largest nodule diameter (mm), (median, range)	80 (10–300)
Number of nodules, *n* (%)	
1	179 (71.9)
2	25 (10.0)
3	8 (3.2)
≥4	37 (14.9)

**Table 5 jcm-11-05802-t005:** Univariate and multivariate analyses.

Variables	Kaplan-Meier		Multivariate Cox Regression Model					
	OS	RFS	OS			RFS		
			HR	95% CI	*p*	HR	95% CI	*p*
Preoperative Imaging								
Tumour size								
≤3 cm vs. >3 cm	0.043	0.187	0.190	0.041–0.878	0.034			
≤5 cm vs. >5 cm	<0.001	0.480	-	-	ns			
Milan criteria (yes vs. no)	<0.001	0.777	-	-	ns			
Portal vein invasion (yes vs. no)	<0.001	0.090	-	-	ns			
Unifocal HCC (yes vs. no)	0.040	0.036	-	-	ns	-	-	ns
Multifocal (yes vs. no)	0.001	0.001	-	-	ns	0.529	0.293–0.954	0.034
Pathological examination								
Tumour size								
≤3 cm vs. >3 cm	0.006	0.182	5.951	1.175–30.150	0.031			
≤5 cm vs. >5 cm	<0.001	0.640						
Unifocal HCC (yes vs. no)	<0.001	0.001	1.775	1.042–3.022	0.035	2.685	1.569–4.593	<0.001
Multifocal HCC (yes vs. no)	0.001	0.018	-	-	ns	-	-	ns
T-Stage	<0.001	0.114	-	-	ns			
G-Stage	<0.001	0.130	-	-	ns			
R-Stage	0.003	0.319	-	-	ns			
V-Stage	<0.001	0.012	1.626	1.093–2.418	0.016	1.643	1.232–2.191	0.001
Desmet	0.603	0.662						
Portal vein invasion	<0.001	<0.001	-	-	ns	-	-	ns
General/surgical parameters								
Age	0.089	0.114						
Sex (male vs. female)	0.121	0.019				1.599	1.004–2.546	0.048
ASA Classification	0.011	0.291	-	-	ns			
Major resection (yes vs. no)	<0.001	0.096	-	-	ns			
Extrahepatic resection (yes vs. no)	0.049	0.019	-	-	ns	-	-	ns
Preoperative laboratory values								
Albumin ^1^	<0.001	0.248	0.414	0.241–0.712	0.001			
Bilirubin ^2^	0.052	0.073						
Quick ^3^	0.167	0.259						
AFP ^4^	0.001	0.046	1.718	1.016–2.906	0.043	-	-	ns

Compared parameters: ^1^ ≤34, >34, ^2^ ≤1.2, 1.21–1.49, >1.49, ^3^ ≤80, >80, ^4^ ≤8.8, >8.9. Perioperative deaths (*n* = 16) were excluded; for multivariate analysis, *p* values < 0.05 were further analysed using backward selection. ns: not significant; OS: overall survival; RFS: intrahepatic recurrence-free survival; HR: hazard ratio; CI: confidence interval; HCC: hepatocellular carcinoma; ASA: American Society of Anaesthesiologists; AFP: alpha-fetoprotein.

**Table 6 jcm-11-05802-t006:** OS regarding to pathological examination.

Variables	*n*	OS			
		Median (mon)	1-year	3-year	5-year
R-Stage					
R0	204	59.9	78%	69%	49%
R1/R2	24	16.3	53%	40%	28%
T-Stage					%
T1	89	85.0	97%	84%	66%
T2	80	46.5	88%	66%	43%
T3	53	28.2	65%	39%	25%
T4	10	9.5	37%	12%	12%
Vascular invasion					
V0	127	72.7	90%	74%	57%
V1	82	41.4	80%	58%	33%
V2	22	19.6	61%	30%	30%

**Table 7 jcm-11-05802-t007:** RFS regarding to pathological examination.

Variables	*n*	RFS			
		Median (mon)	1-year	3-year	5-year
R-Stage					
R0	204	11.8	49%	20%	6%
R1/R2	24	8.4	29%	21%	14%
T-Stage					
T1	89	24.7	62%	32%	9%
T2	80	12.4%	51%	16%	4%
T3	53	8.7%	31%	15%	8
T4	10	7.7%	22%	11%	11
Vascular invasion					
V0	127	18.0%	57%	26%	8%
V1	82	10.4%	42%	17%	5%
V2	22	7.3%	18%	6%	6%

## Data Availability

Not applicable.

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
