# Peer review of "Outcome after Resection for Hepatocellular Carcinoma in Noncirrhotic Liver—A Single Centre Study"

_jcm, 2022, doi:10.3390/jcm11195802_

Round 1
Reviewer 1 Report
Although this is a single-center study, it is a detailed and well-written paper. It seems that they are actively performing surgeries in difficult situations, such as patients in poor general condition and those with distant metastases. I have a few questions, which I will address below.
The authors state that "all treatment decisions were made by a multi-disciplinary tumour board”. This study includes multiple tumors and M1 cases for which resection is not recommended by recent guidelines. Since no resection criteria are listed in the text, the criteria at the institution should be clearly stated.
In some Tables, the total does not add up to 247. It is necessary to confirm that the numbers are accurate.
Regarding Table 5, the description of each factor is not sufficient. For example, in the Sex section, is the risk higher for males or females? Which has a worse prognosis, Unifocal or Multifocal? I am not a statistician, so if the Table is statistically correct, no correction is necessary.
Does Desmet mean fibrosis score?
It would be easier to understand if "%" was added to Table 7 as in Table 6.
Reviewer 2 Report
Authors evaluated this retrospective study that the outcomes of resections for NC-HCC and to determine prognostic factors for overall survival and intrahepatic recurrence-free survival (OS, RFS). They exposed that tumor diameter greater than three centimeters, multifocal tumor disease, vascular invasion, preoperative low albumin and increased alpha-feto-protein (AFP) values were associated with significantly worse OS in patients underwent liver resection with NC-HCC.
I would like to REVIEW of this manuscript with very interesting. However, it had several concerns.
Major
Authors should be described about detail of study design
1. Surgical indication was not clear. How did authors determine preoperative liver function? There have been 5 patients with acute liver failure after major resection, I concerned that evaluation of preoperative liver function and indication of surgery might be poor.
2. I think that a total of 16 patients (6.4%) died within hospital stay is a lot. Please state your reasons.
3. Authors had 12 patients which were underwent preoperative treatment such as RF and TACE, should you have excluded from this study? Because those patients might already have some bias in this study.
4. I believe there are so many cases of lymph node metastasis and underwent lymphadenectomy.
5. There is no description of histological classification on the pathology.
6. Are there any new findings compared to previous papers?
Round 2
Reviewer 2 Report
1) Surgical indication was not clear. How did authors determine preoperative liver function? There have been 5 patients with acute liver failure after major resection, I concerned that evaluation of preoperative liver function and indication of surgery might be poor.
Response:
Thank you very much for your helpful comment. For the first part concerning the indication for liver resection we would like to refer to our statement on reviewer 1’s first comment.
As for the second part of your comment, we determined liver function preoperatively in the first step via laboratory values. Only patients with unimpaired liver function qualified for liver resection. In a second step, preoperative imaging was examined for signs of portal hypertension. Patients with splenomegaly, intraabdominal varices or ascites were excluded. In case of extended resections, volumetry based on current imaging was performed by a radiologist specialised in visceral imaging in case of doubt for estimation of the future liver remant.
We added following section to the Material and Methods section:
“Only patients with unimpaired liver function qualified for resection and those with signs of portal hypertension like splenomegaly, intraabdominal varices or ascites were no candidates for resection. If extended resections were necessary additional volumetry of the future liver remnant was performed by an experienced radiologist.”
>> The description of the resection criteria is still unclear; Is it not stated whether Child-Pugh or Makuuchi criteria were used? Also, I understood that volumetry was performed, however
there is no mention of how much remnant liver volume was required for major liver resection.
If the remnant liver volume is low In CT volumetry, is the preoperative portal vein embolization not required?
3) Authors had 12 patients which were underwent preoperative treatment such as RF and TACE, should you have excluded from this study? Because those patients might already have some bias in this study.
Response:
You are of course right that some bias is created by the inclusion of preoperatively treated patients. However, we have decided to include these patients in the study, too, as these patients are also part of the surgical collective. Besides, excluding them would have meant to create another kind of bias.
Overall, multimodal therapy of HCC is gaining more and more importance. For example, surgical therapy can also serve as a second step after TACE has been performed for a kind of bridging and estimation of tumor biology.
>>I guess, In case of Including preoperative treatments does not reflect the pure efficacy of resected treatment.
Author Response
Comment 1
Thank you very much for your inquiry. In principle, 5ml of healthy liver tissue per kilogram of body weight is considered the absolute minimum at our center. In case of doubt, the upper limit of the resection extent is determined in this way.
Only patients with HCC without the presence of cirrhosis were included in this study. Therefore, Child-Pugh classification, which requires the presence of cirrhosis, was not used. However, absolute exclusion criteria for resection of HCC in general in our institution are signs of portal hypertension on preoperative imaging or decompensated liver function by means of laboratory values (i.e. bilirubin, INR, albumin) or the presence of ascites.
Regarding partial portal vein occlusion as a hypertrophic stimulus, we are very cautious about HCC at our center. Portal vein embolization is an option in borderline cases, but it was not necessary in any of our patients. In addition, we do not consider HCC - whether in cirrhosis or non-cirrhosis - to be a good indication for ALPPS.
Comment 3
Thank you for your thoughtful comment, you raise an important point here. We respect your opinion that preoperatively treated HCC no longer reflect the pure effectiveness of surgical therapy.
Nevertheless, we believe that we should not exclude preoperatively treated patients from the collective. We felt it was important to show that HCC treated preoperatively by, for example, TACE or RFA does not exclude resection in a second step. Moreover, postoperative follow-up treatment for HCC is now the standard and no longer the exception. These HCC are also included in the analysis of overall survival. Therefore, a 'purely surgical' outcome often cannot be considered.